# Effects of the INDC and GGRMA Regulations on the Impact of PM$_{2.5}$ Particle Emissions on Maritime Ports: A Study of Human Health and Environmental Costs

**Ching-Chih Chang [1,*], Yu-Wei Chang [2] and Po-Chien Huang [2]**

[1] Department of Transportation and Communication Management Science, The Research Center for Energy Technology and Strategy, National Cheng Kung University, No. 1, University Road, Tainan 70101, Taiwan

[2] Department of Transportation and Communication Management Science, National Cheng Kung University, No. 1, University Road, Tainan 70101, Taiwan; wade.chang@asml.com (Y.-W.C.); 1011238103@stu.nkmu.edu.tw (P.-C.H.)

[*] Correspondence: chan5305@mail.ncku.edu.tw

**Abstract:** This study employs an activity-based model to estimate the PM$_{2.5}$ particle emissions from ships, cargo-handling equipment, and heavy-duty vehicles in the Port of Kaohsiung, Taiwan. External health costs, the index of health impact (IHI), and external environmental costs are assessed to quantify the impact of PM$_{2.5}$ particle emissions. The mitigation regulations applied in this study are the Intended Nationally Determined Contribution Act (INDC) and the Greenhouse Gas Reduction and Management Act (GGRMA). The provisions in these acts are incorporated into Scenario-INDC and Scenario-GGRMA. The results are as follows: from 2005 to 2017, PM$_{2.5}$ particle emissions caused an external health cost of 3238.30 DALY (disability-adjusted life year), an IHI value of 8.53%, and environmental cost of USD 2176.04 million annually. For Scenario-INDC and Scenario-GGRMA, it is predicted that PM$_{2.5}$-related external health costs, IHI value, and external environmental cost will decrease by 927.64 DALY, 2.45%, and USD 608.86 million and by 1736.28 DALY, 4.58%, and USD 1139.84 million, respectively, as compared to BAU-2030 and BAU-2050. The results indicate that compliance with INDC and GGRMA regulations will lead to a significant mitigation of PM$_{2.5}$ particle emissions, resulting in a significant improvements in air quality and human health in addition to a reduction in environmental costs.

**Keywords:** PM$_{2.5}$ particles; environmental; health; transportation; pollution

## 1. Introduction

The World Health Organization (WHO) [1] points out that from 2010 to 2016, only 18% of the global population lived in an environment that met the Air Quality Guidelines of WHO (AQG: a PM$_{10}$ concentration of 20 μg/m$^3$, PM$_{2.5}$ concentration of 10 μg/m$^3$). The population of those exposed to long-term high concentrations of PM$_{10}$ and PM$_{2.5}$ particles increased during that period of time: Residents living in PM$_{10}$ concentrations of 30 μg/m$^3$ and PM$_{2.5}$ concentrations of 15 μg/m$^3$ increased by 31%; those living in PM$_{10}$ concentrations of 50 μg/m$^3$ and PM$_{2.5}$ concentrations of 25 μg/m$^3$ increased by 50%; and those living in PM$_{10}$ concentrations of 70 μg/m$^3$ and PM$_{2.5}$ concentrations of 35 μg/m$^3$ increased by 63%. European Environment Agency (EEA) [2] data show that in 2015, approximately 13% of PM$_{2.5}$ and PM$_{10}$ emissions in the EU came from the transportation sector. Furthermore, due to prosperous international trading, shipping plays an important role in this issue; hence, its effect on air pollution should not be ignored.

Numerous studies have reported long-term exposure to PM$_{2.5}$ to be associated with abnormalities of the lung and cardiovascular systems, which can lead to an increased likelihood of mortality and morbidity from stroke, ischemic heart disease (IHD), lung cancer, chronic obstructive pulmonary disease (COPD), and acute lower respiratory infections

(ALRIs). In addition, long-term exposure to $PM_{2.5}$ appears to have an effect on children's neurodevelopment and cognitive abilities [2–10]. According to the WHO [1], approximately 7 million deaths can be attributed to exposure to particulate matter globally each year. Furthermore, higher concentrations of $PM_{2.5}$ in the living environment lead to higher risks and mortality from $PM_{2.5}$-related diseases [11,12]. Yang and Kao [3] conducted a study on the association between $PM_{2.5}$ concentration and mortality in Taiwan, and the results showed that if concentrations of $PM_{2.5}$ in Taiwan decreased to 15 $\mu g/m^3$ [13] and 10 $\mu g/m^3$ (the annual standard of the World Health Organization), total deaths would decrease by 6.7% and 8.4%, respectively, and the life expectancy of the population would increase due to improved air quality.

In addition to the negative impacts of related diseases or death, fine particulate matter has economic costs for society as well. Gu et al. [14] and Chen et al. [15] concretized the economic losses due to $PM_{2.5}$ in China. These studies indicated that from 2014 to 2016, the impact of $PM_{2.5}$ pollution reduced the GDP of urban areas by around 0.3% to 1%. Chatzinikolaou et al. [16] assessed the local impact of $PM_{2.5}$ and $PM_{10}$ emissions in the Port of Piraeus in Greece. Their results showed that $PM_{2.5}$ and $PM_{10}$ emissions led to approximately EUR 26,314,700 in economic damage during the study period, and they found that the pollutant that contributed to the most damage was $PM_{2.5}$. The per capita external cost of $PM_{2.5}$ within 56 km of the port area was EUR 4, while the per capita external cost within 300 km of the port area was approximately EUR 1.7.

The International Energy Agency (IEA) [17] announced that in 2017, energy usage in the global transportation sector accounted for 29% of total energy usage, of which cargo transportation (including road, rail, maritime, and air) accounted for 40% of transportation energy consumption. Although the transportation sector is not a major factor in economic and social development, the negative impact of the transportation sector on the environment cannot be ignored. European Environment Agency (EEA) [2] data showed that in 2015, approximately 13% of the fine particulate matter and particulate matter in the main sources of emissions in the EU came from the transportation sector. Moreover, due to rapid growth in international trading, shipping plays an important role in international transportation, and its effect on air pollution should thus not be ignored. Nickel (Ni) and vanadium (V) elements in maritime-related $PM_{2.5}$ are closely related to $PM_{2.5}$-related diseases [18], especially respiratory disease, cardiovascular disease, and lung cancer [6,19–21].

According to the Environmental Protection Agency [22] of Taiwan (the legal standard for the annual average $PM_{2.5}$ concentration is 15 $\mu g/m^3$), only 12% of Taiwan's counties are meeting these standards; 31% are higher than 25 $\mu g/m^3$, and 25% are higher than 20 $\mu g/m^3$. This indicates that the population in most of Taiwan is at risk for health-related problems due to emissions [23]. Furthermore, between 2005 and 2017, Kaohsiung City was the region with the highest annual $PM_{2.5}$ emissions in Taiwan, and a large part of these emissions were due to maritime international trading and related land transportation [22].

As of now, there is no formal emission mitigation goal for $PM_{2.5}$ in Taiwan. However, in 2015, the government of Taiwan did pass two pieces of greenhouse emissions legislation: the Intended Nationally Determined Contribution Act [24] and the Greenhouse Gas Reduction and Management Act [25]. The INDC and GGRMA are aimed at reducing the level of greenhouse gas emissions measured in 2005 by 20% by 2030 and by 50% by 2050. To see how achieving these goals might affect the economy and health of the region, an activity-based model is used to analyze $PM_{2.5}$ emissions from shipping-related transportation and to further assess the external environmental costs and health costs in Kaohsiung, which is where the largest port in Taiwan is located [15,26–28].

This study is organized into four stages. In the first and second stages, $PM_{2.5}$ emissions in 2030 and 2050 are projected without the application of regulations required by the INDC and the GGRMA, where these results are described as business as usual (BAU). In the third and fourth stages, it is assumed that the required INDC and GGRMA regulations have been met, and the impact this would have on $PM_{2.5}$ emissions and in turn on environment and health conditions in Kaohsiung are compared to those in 2005.

Based on the discussion above, the purposes of this study are as follows:

1.　Estimate the health and environmental costs from the $PM_{2.5}$ emissions from shipping-related transportation involved in international trading in Kaohsiung from 2005 to 2017;

2.　Estimate the health and environmental costs from the $PM_{2.5}$ emissions from shipping-related international trading transportation in Kaohsiung in 2030 and 2050 in a business as usual (BAU) scenario, where none of the regulations specified in the INDC or GGRMA are applied;

3.　Estimate the reduction in $PM_{2.5}$ emissions and the projected health and environmental costs from the $PM_{2.5}$ emissions from shipping-related international trading transportation in Kaohsiung in 2030 and 2050 when the regulations specified in the INDC or GGRMA are applied (Scenario-INDC and Scenario-GGRMA).

The contribution of this study is to estimate the projected $PM_{2.5}$ emissions if the goals of INDC and GGRMA were to be achieved, how much the external environmental cost of $PM_{2.5}$ emissions would be including health costs, and the percentage of decrease (IHI value) that would occur in related diseases and environmental costs. The decreases in the IHI value would prevent $PM_{2.5}$-related diseases. These diseases include stroke, ischemic heart disease, lung cancer, chronic obstructive pulmonary disease, acute lower respiratory infections, etc.

## 2. Research Methodology

### 2.1. Data Desciption

The research data analyzed in this study can be divided into three types of shipping transportation data: ships, cargo-handling equipment, and heavy-duty vehicles (Table 1). The data source is the Annual Statistical Report of the Port of Kaohsiung [29,30]. Figure 1 shows the research boundary of Port of Kaohsiung.

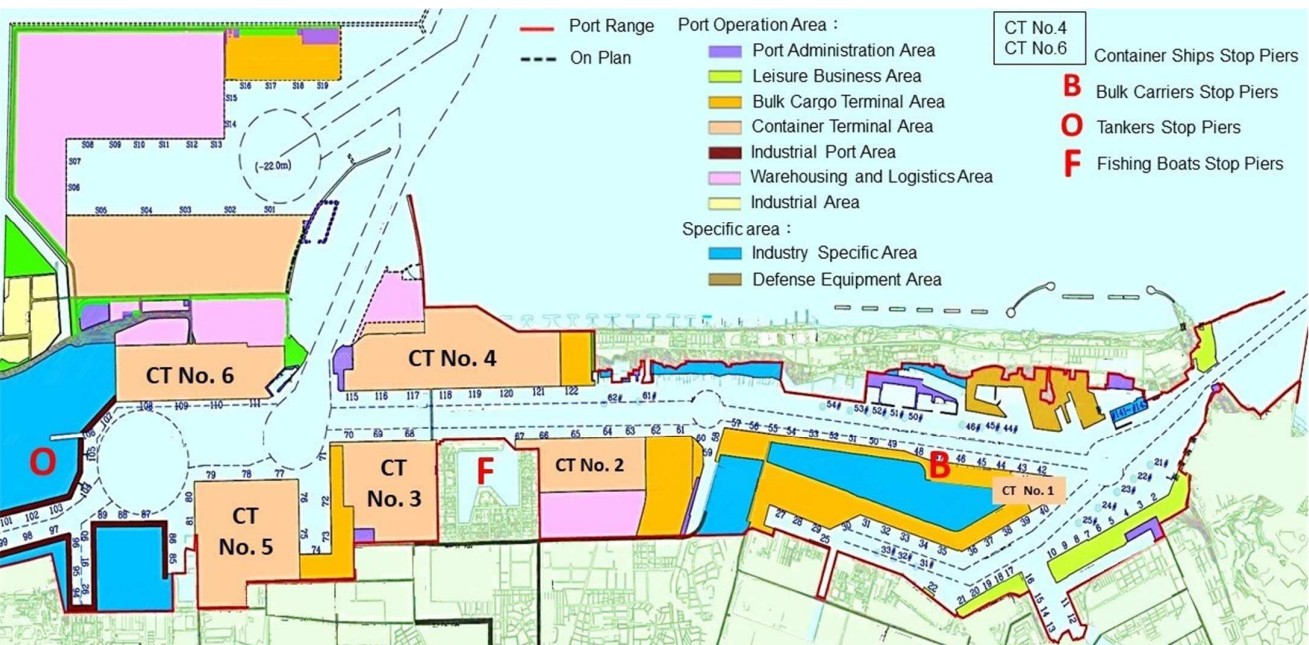

**Figure 1.** Research Boundary of Port of Kaohsiung.

**Table 1.** The basic data of the research object.

| Objects | Description |
|---|---|
| Ship | Types of ships: container ships, bulk ships, tankers, and others. A total of 18,773 ships entered into Kaohsiung port in 2017, including 9349 container ships, 5161 bulk carriers, 2992 tankers, and 1271 other ships. |
| Cargo-Handling Equipment | Take gantry cranes as the research object. A total of 68 gantry cranes are included in the port of Kaohsiung. |
| Heavy Vehicles | Take tractors as the research objects. A total of 544 tractors are included in the port of Kaohsiung. |

*2.2. Notation*

This study takes data related to shipping-related transportation for international trading in Kaohsiung as the basis for model building. Descriptions of the index, variables, and parameters in this study are shown in Table 2.

**Table 2.** Notation of index, variables, and parameters.

| Index | Description | |
|---|---|---|
| $i$ | $i = 1\sim3$ (1 = Ships, 2 = Cargo-handling equipment, 3 = Heavy-duty vehicles) | |
| $j$ | $j = 1\sim4$ (1 = Container ships, 2 = Bulk ships, 3 = Tankers, 4 = Other ships) | |
| $k$ | $k = 1\sim3$ (1 = Cruising, 2 = Maneuvering, 3 = Hotelling) | |
| $l$ | $l = 1\sim4$ (1 = Container, 2 = Bulk, 3 = Oil cargo, 4 = Fish cargo) | |
| $E_i$ | PM$_{2.5}$ emissions from source $i$ (1 = Ship, 2 = Cargo-handling equipment, 3 = Heavy-duty Vehicle) | ton |
| $EC_{Health}$ | External environmental health costs | DALY |
| $EC_{Environmental}$ | External environmental costs | USD/ton |
| Energy | Energy consumption of ships | kW-h |
| $EF_i$ | Emission source $i$ | g/kW-h (heavy-duty vehicle in term of g/mile) |
| $FCF_i$ | Fuel correction emission source $i$, corrected based on different fuels | No unit |
| $N_j$ | Number of ships of type $j$ | fleet |
| MCR | Maximum continuous rated engine power | kW |
| $LF_j$ | Load factor for ship type $j$ | No unit |
| $Act_i$ | Activity of emission source $i$ | h (heavy-duty vehicle in term of mile) |
| $AS_{jk}$ | Actual speed of type $j$ ship on an activity $k$ | knot |
| $MS_j$ | Maximum speed of ship type $j$ | knot |
| $D_k$ | Distance of ship on activity $k$ | mile |
| HP | Rated horsepower for cargo-handling equipment | hp |
| ZH | Zero-hour emission rate | g/hp·h |
| DR | Deterioration rate | g/hp-h$^2$ |
| CommHours | Cumulative hours of cargo-handling equipment | h |
| $N_i$ | Amount of emission source equipment type $i$ | |
| $Q_l$ | Handling cargo type $l$ | ton |
| $HV_l$ | Maximum load weight of heavy-duty vehicles of cargo type $l$ | ton |
| $X_t$ | Forecasting value at period $t$ | fleet, TEU, ton |
| $Y_t$ | Actual value at period $t$ | fleet, TEU, ton |
| $\alpha$ | Smoothing constant, $0 < \alpha < 1$ | No unit |
| n | Actual numerical and forecasting values | year |
| $A_t$ | Actual value at year $t$ | fleet, TEU, ton |
| $E_t$ | Forecasting value at year $t$ | fleet, TEU, ton |
| $factor_H$ | External environmental health cost | DALY/ton |
| $GBD_{PM2.5}$ | Global burden of disease of PM$_{2.5}$-related disease | DALY |
| $Pop_1$ | Size of population | person |
| $Pop_2$ | Population burden of disease, based on 500,000 people | person |
| $factor_E$ | External environmental costs | USD/ton |

*2.3. Models*

2.3.1. Model of PM$_{2.5}$ Emission

The top-down approach and the bottom-up approach (also known as activity-based model) are the two main methodologies for evaluating emissions. The top-down approach estimates emissions based on the usage of fuel, whereas the bottom-up approach estimates emissions based on the research target's activities. In general, a bottom-up approach is usually more accurate than the top-down approach [31,32].

In this study, the bottom-up approach (activity-based method) is used to estimate the emissions from ships, cargo-handling equipment, and heavy-duty vehicles [28,33]. The Puget Sound Maritime Air Emissions Inventory [34], POLA [35–37], and San Pedro Bay [38] 17 are used to estimate emissions in Kaohsiung. This study only takes ship activities within 40 nautical miles of the Port of Koahsiung into account. The activities of ships included in the study can be divided into three phases: (1) cruising phase (ship activities within 40 miles of the entrance of the port); (2) maneuvering phase (ship activities from the entrance of port to the berth); and (3) hotelling phase (ships stopped at their berths). The emissions model for the ships in port is shown in Equations (1)–(3).

$$E_1 = \text{Energy} \times \text{EF}_1 \times \text{FCF}_1 \times N_j \times 10^{-6} \tag{1}$$

$$= \text{MCR} \times \text{LF}_j \times \text{Act}_1 \times \text{EF}_1 \times \text{FCF}_1 \times N_j \times 10^{-6} \tag{2}$$

$$= \text{MCR} \times \left(\frac{\text{AS}_{jk}}{\text{MS}_j}\right)^3 \times \left(\frac{D_k}{\text{AS}_{jk}}\right) \times \text{EF}_1 \times \text{FCF}_1 \times N_j \times 10^{-6} \tag{3}$$

For cargo-handling equipment, this study refers to the emissions model for cargo-handling equipment based on Puget Sound Maritime Emissions Inventory [34], POLA [35,36], and Futaba [39]. The factors in this model are shown in Equations (4) and (5).

$$E_2 = \text{HP} \times \text{LF}_2 \times \text{Act}_2 \times \text{EF}_2 \times \text{FCF}_2 \times N_2 \times 10^{-6} \tag{4}$$

$$= \text{HP} \times \text{LF}_2 \times \text{Act}_2 \times (\text{ZH} + \text{DR} \times \text{CommHours}) \times \text{FCF}_2 \times N_2 \times 10^{-6} \tag{5}$$

For the analysis of heavy-duty vehicles in and out of the port, Puget Sound Maritime Emissions Inventory [34] and POLA [35] are used in the current study to build an emission model for heavy-duty land transportation vehicles, as shown in Equations (6) and (7).

$$E_3 = \text{EF}_3 \times \text{Act}_3 \times N_3 \times 10^{-6} \tag{6}$$

$$= \text{EF}_3 \times \text{Act}_3 \times \left(\frac{Q_l}{\text{HV}_l}\right) \times 2 \times 10^{-6} \tag{7}$$

2.3.2. Forecasting Model

This study makes use of data related to the forecasting of cargo handling reported by the Annual Statistical Report, Port of Kaohsiung [30]. On the basis of the exponential smoothing model [40,41], trend forecasting and seasonality are taken into consideration to estimate the number of vessels entering into the Port of Kaohsiung and the volume of cargo that will be handled. The forecasting model is shown as Equation (8).

$$X_{t+1} = X_t + \alpha(Y_t - X_t) = \alpha Y_t + (1 - \alpha)X_t \tag{8}$$

The mean absolute percentage error (MAPE) is used to assess the accuracy of the forecasting value of the exponential smoothing model, as shown in Equation (9).

$$\text{MAPE} = \frac{\sum_{t=1}^{n} \left(\frac{A_t - E_t}{A_t}\right)}{n} \times 100\% \tag{9}$$

2.3.3. External Environmental Costs

Environmental costs can be divided into two parts: The first part comprises external health costs and the index of health impacts, and the second part comprises the external environmental costs. For the disability-adjusted life year (DALY) of external health costs, this study follows the research results of ITSUBO and INABA [42], Tang et al. [43], and Kyu et al. [44], and the model is further modified, as shown as Equation (10).

$$EC_{\text{Health}} = E_i \times factor_{\text{H}} \tag{10}$$

This study refers to a study of the global cost of diseases [44] to define $PM_{2.5}$-related diseases. The model for assessing the impact of air pollutant-disease is shown as Equation (11).

$$\text{IHI} = \left( \frac{EC_{\text{Health}}}{\text{Pop}_1} \Big/ \frac{GBD_{\text{PM2.5}}}{\text{Pop}_2} \right) \times 100\% \tag{11}$$

The second step for estimating environmental costs is to assess the environmental impact of $PM_{2.5}$ emissions [26,45,46]. The modified model of external environmental costs is shown as Equation (12).

$$EC_{\text{Environmental}} = E_i \times factor_{\text{E}} \tag{12}$$

## 3. Empirical Results

In this section, we first estimate the $PM_{2.5}$ emissions and external environmental costs for shipping-related transportation in Kaohsiung from 2005 to 2017. Then, we predict the entry of ships and the volume of cargo handling in the Port of Kaohsiung. The third step is an analysis of emissions in Kaohsiung using four different scenarios. Finally, we explore the health and environmental impacts of emissions in Kaohsiung under four different scenarios.

### 3.1. Data Analysis for Shipping-Related Emissions in Kaohsiung

In this study, the number of ships entering the port of Kaohsiung from 2005 to 2017 is taken as the database from which to estimate the $PM_{2.5}$ emissions of the ships in port. The results are shown in two ways: types of activity and ship category. In terms of activity type, the main type of ship activity is hotelling, which accounts for 74% of total ship emissions, followed by cruising (24%) and maneuvering (2%). Although the average emissions at any particular time period from cruising and maneuvering are higher than those for the time spent in berth, the emissions during berthing are considerably greater than those of the other two phases because of the time spent in berth. In terms of ship type, the emissions from container ships was the largest during the study period, with average annual $PM_{2.5}$ emissions of approximately 1953.65 tons/year, which accounted for 71% of total ship emissions. Other ship categories include bulk ships (421.70 emission tons/year), tankers (264.23 emission tons/year), and fishing ships (101.41 emission tons/year), which account for 15%, 10%, and 4% of emissions, respectively. As can be seen, reducing the emissions of container ships would significantly decrease total $PM_{2.5}$ emissions.

Gantry cranes are the main cargo-handling equipment (CHE) source of emissions. The annual emissions from gantry cranes are approximately 126.78 tons, which accounts for 59% of the emissions of all cargo-handling equipment. The second-highest emitter of $PM_{2.5}$ are RTG Cranes, with annual emissions of 84.80 tons, representing 39% of the total CHE emissions. Container forklifts emit 3.76 tons of $PM_{2.5}$, which is 2% of the total. Among the container forklifts, the annual $PM_{2.5}$ emissions of laden container forklifts is 0.96 tons, while that of empty container forklifts is 2.80 tons.

From 2005 to 2017, the main source of emissions from heavy-duty vehicles in the port was from container tractors, which accounted for 64.85% of the total HDV emissions in the port, and the next-highest emissions sources during the study period were bulk trucks (34.68%), tank trucks (9.56%), and fish trucks (0.36%). If heavy-duty vehicles outside the port are taken into consideration, the proportion of the emissions of container tractors

would decline by 20.24%, meanwhile those of bulk trucks, tank trucks, and fish trucks would increase by 6.05%, 4.56%, and 0.06%. When all land transportation is taken into consideration, the emissions from container tractors and bulk trucks are almost the same, where these emissions account for the smallest proportion of total HDV emissions. Generally, the percentage of the emissions due to land transportation in the port and out of the port were 9.50% and 90.5%, respectively. This means that out of the port emissions are the main source of the heavy-duty vehicle emissions.

Table 3 shows the contributions of the emission sources. The contribution of emissions from ships, cargo-handling equipment, and heavy-duty vehicles to total emissions is 84%, 7%, and 9%, respectively. The highest proportion of emissions comes from ships, and the lowest contribution comes from cargo-handling equipment. The highest percentage of ship types entering the Port of Kaohsiung are container ships (60.20%), followed by bulk ships (13.00%) and tankers (8.14%). In the case of cargo-handling equipment, the proportion of emissions from gantry cranes and RTG cranes accounts for approximately 6.52% of total emissions, while container forklifts (including laden container forklifts and empty container forklifts) account for up to 0.12% of total emissions. In terms of heavy-duty vehicles, the emissions from container tractors and bulk trucks are greater than those from other heavy-duty vehicles. Altogether, heavy-duty vehicles account for approximately 7.59% of the total emissions. Because of the huge emissions from ships, other emissions sources are relatively small. However, the annual emissions due to cargo-handling equipment and heavy-duty vehicles are 215.35 tons and 288.69 tons, respectively, which, on its own, has a great impact on the environment and on local residents. Therefore, these external costs cannot be ignored.

**Table 3.** The percentage of emission sources (by type of ship, CHE, and HDV).

| Emission Source | Type | Contribution | Total |
|---|---|---|---|
| Ship ($E_1$) | Container Ship | 60.20% | 84.47% |
| | Bulk Ship | 13.00% | |
| | Tanker | 8.14% | |
| | Other Ship (Fish Ship) | 3.13% | |
| Cargo-Handling Equipment ($E_2$) | Gantry Crane | 3.91% | 6.64% |
| | RTG Crane | 2.61% | |
| | Laden Container Forklift | 0.03% | |
| | Empty Container Forklift | 0.09% | |
| Heavy-Duty Vehicle ($E_3$) | Container Tractor | 3.97% | 8.90% |
| | Bulk Cargo Truck | 3.62% | |
| | Tank Truck | 1.26% | |
| | Fish Truck | 0.05% | |

*3.2. Forecasting of Fleets Entering the Port of Kaohsiung and the Amount of Cargo Handling*

3.2.1. MAPE

In this study, MAPE is used as the methodology by which to assess the error rate in the forecasting values for four types of ships. The results indicate that the error rates for the forecasts of container ships, bulk ships, tankers, and other ships are 7.97%, 3.90%, 4.32%, and 5.68%, respectively, with an average error rate of 5.47%, which is less than 10%. This means that the forecasting is highly accurate. In this study, the cargo volume handled in the Annual Statistical Report Port of Kaohsiung in 2030 and 2050 is also forecast based on data from the Annual Statistical Report Port of Kaohsiung from 2005 to 2017. The results show that the forecasting error rate for these predictions related to the handling of container, bulk, oil, and ship cargoes were 7.16%, 6.92%, 3.49%, and 19.78%, respectively, during the period under consideration, with an average value of less than 10%, indicating that the forecasting is highly accurate.

### 3.2.2. Forecasting Future Fleets in the Port of Kaohsiung

According to the results of the forecasting model, the proportion of the four types of ships forecast to enter the Port of Kaohsiung in 2030 and 2050 are similar: The fleets of container ships account for 50%, and bulk ships, tankers, and fishing ships account for 30%, 16%, and 7%, respectively. Although the number of ships entering the Port of Kaohsiung in any two given years will be different, the proportion of ships will be similar in the future.

### 3.2.3. Quantity of Handling Cargoes

The amount of cargo projected to be handled in 2030 and 2050 is shown in Table 4. It can be seen that there are more handling containers in 2030 and 2050 than in 2005. The volume of bulk and oil cargo in 2030 and 2050 are less than that in 2005. Fish cargoes are greater in 2030 than in 2005 and 2050.

**Table 4.** The forecast of the value of cargoes handled in 2030 and 2050.

| Year | Container (TEU) | Bulk Cargo (ton) | Oil Cargo (ton) | Fish Cargo (ton) |
|---|---|---|---|---|
| 2005 | 9,471,056 | 70,177,402 | 22,889,609 | 660,146 |
| 2030 | 10,193,470 | 54,316,146 | 17,424,652 | 759,121 |
| 2050 | 10,621,486 | 51,554,801 | 16,544,681 | 576,988 |

### 3.3. Shipping-Related Emissions under Different Scenarios

Based on the forecasts of the number of ships and the volume of cargo handled, an estimate of the $PM_{2.5}$ emissions due to shipping-related international trading transportation in Kaohsiung is derived. As shown in Figure 2, the emissions in 2030 and 2050 are projected to be 3446.86 tons and 3290.39 tons, respectively. Compared with 2005 (3152.46 tons), emissions in the future will increase if no mitigation regulations are applied. If the regulations included in the Intended Nationally Determined Contribution Act are observed, emissions will decrease by 2030 to 2521.97 tons, which is 80% of the volume in 2005. Emissions will be reduced by 924.88 tons in 2030 and 1714.16 tons in 2050 in comparison with the emissions in business as usual scenarios if the government's emission mitigation goals are met.

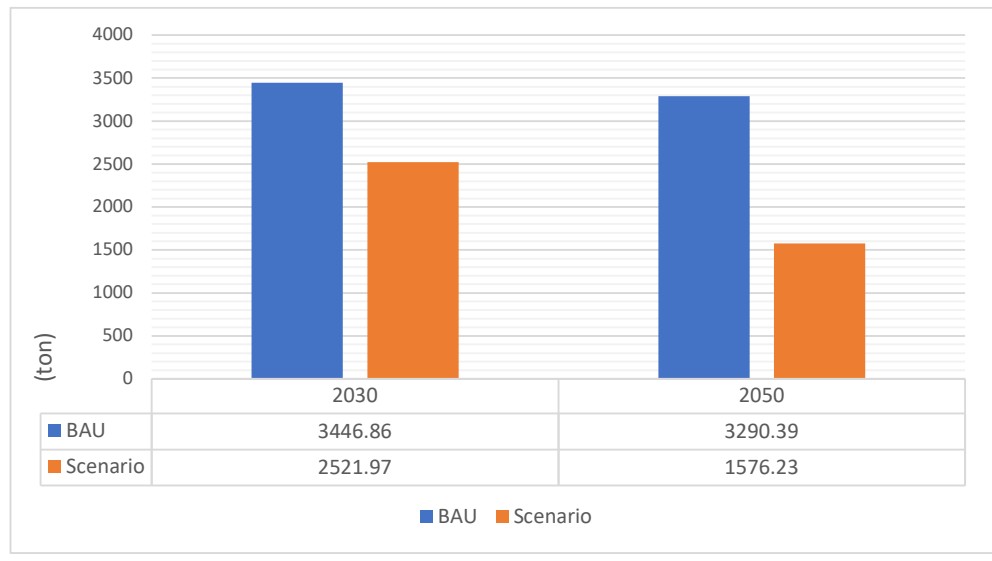

**Figure 2.** The emissions of shipping-related transportation under different scenarios.

### 3.4. External Environmental Costs of Shipping-Related Emissions

3.4.1. The External Environmental Costs Derived from the Basic Data

This study refers to the damage factor suggested by Tang et al. [43] for air pollutants and takes the disability-adjusted life years (DALYs) as the unit to quantify the local health

impact from PM$_{2.5}$ emissions due to shipping-related transportation. The World Health Organization defines the disability-adjusted life year, as shown in Figure 3, as a quantitative method for measuring the cost of disease. It is commonly applied in the area of health assessment. In addition to life-year loss due to illness, the life-years of disease damage and disability are also taken into consideration. When calculating the health cost of a disease, the years of potential life lost (YLL) is used to indicate the years of potential life lost due to a given disease. In addition, the measure includes years where life is not actually lost, but health is impaired, or a person is disabled due to their disease. The years of life with disease damage or disability are represented as years lived with disability (YLD). The disability-adjusted life years (DALYs) are the sum of the YLL and YLD.

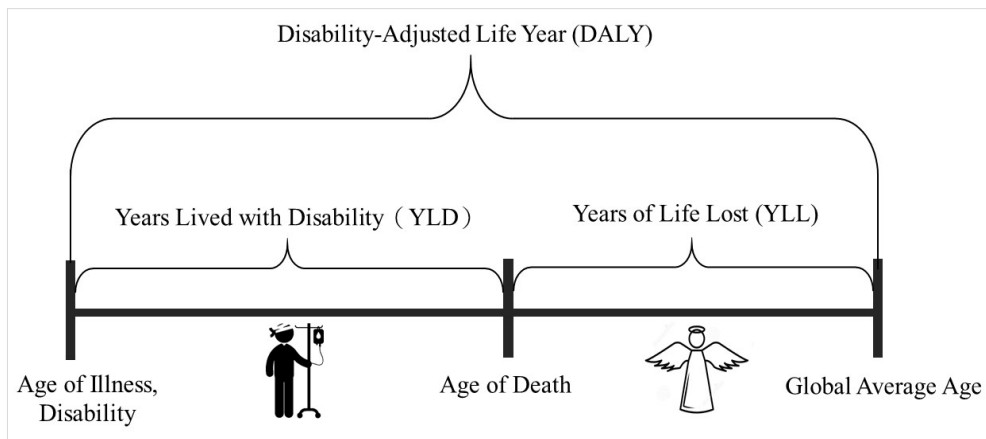

**Figure 3.** The concept for the disability-adjusted life year.

In this study, it is assumed that the PM$_{2.5}$ emissions from shipping-related transportation in Kaohsiung will only affect the local residents and environment. The external environmental costs of PM$_{2.5}$ emissions from 2005 to 2017, including local health and environmental impacts, is assessed. The results are as follows: In terms of external health costs, as shown in Table 5, from 2005 to 2017, the health costs due to shipping-related PM$_{2.5}$ emissions was approximately 2644.92 DALYs per year for residents in Kaohsiung. The external health issues caused by ship emissions was 2299.69 DALYs, which accounts for the majority of all emission sources. The next-largest sources of emissions were heavy-duty vehicles (261.54 DALYs) and cargo-handling equipment (83.94 DALYs). The index of health impact (IHI) value of PM$_{2.5}$ emissions in Kaohsiung from 2005 to 2017 was 7.17%, which means that 7.17% of PM$_{2.5}$-related diseases, including stroke, ischemic heart disease, lung cancer, chronic obstructive pulmonary disease, and acute lower respiratory infections, could have been caused by the PM$_{2.5}$ emissions due to shipping-related transportation in Kaohsiung. The IHI value in 2016 was the highest, which was about 8.09%. In terms of external environmental costs, the annual environmental cost loss in Kaohsiung was USD 1787.32 million from 2005 to 2017. The average annual loss due to ships (USD 1509.70 million) accounted for the highest proportion of this loss, followed by heavy-duty vehicles (USD 159.01 million) and cargo-handling equipment (USD 118.61 million).

**Table 5.** External environmental costs of shipping-related emissions from 2005 to 2017.

|  | Ship | Cargo-Handling Equipment | Heavy-Duty Vehicles | Total |
|---|---|---|---|---|
| External Health Costs (DALYs) | 2299.44 | 83.94 | 261.54 | 2644.92 |
| Index of Health Impact (%) | 6.06 | 0.48 | 0.64 | 7.17 |
| External Environmental Costs (USD million) | 1509.70 | 118.61 | 159.01 | 1787.32 |

### 3.4.2. External Environmental Costs under the Various Scenarios

This study estimates the external environmental costs from shipping-related emissions in 2030 and 2050 based on $PM_{2.5}$ emissions in a BAU scenario. As can be seen from Figure 4, the external health cost in 2030 will be 2891.91 DALYs, which is 246.99 DALYs higher than in 2005. In 2050, the external health cost will be 2760.63 DALYs, which is 115.71 DALYs higher than emissions in 2005. In terms of types of emission sources, the external health costs of ships will still be 2430.53 and 2301.02 DALYs in 2030 and 2050, respectively. The annual health cost of ships is 2292.66 DALYs, which makes this the major proportion of emissions leading up to 2050. Compared with the external health costs of ships, the health cost of cargo-handling equipment and heavy-duty vehicles accounts for smaller portions, which are 214.35 DALYs and 231.83 DALYs, respectively. Without factoring in other external factors, such as acts of nature (e.g., wind), container ships cause the most external health costs. From 2030 to 2050, the annual external health costs of container ships will be 1650.55 DALYs, while costs from bulk ships, tankers, and fishing ships will be 342.05, 221.78, and 78.28 DALYs, respectively.

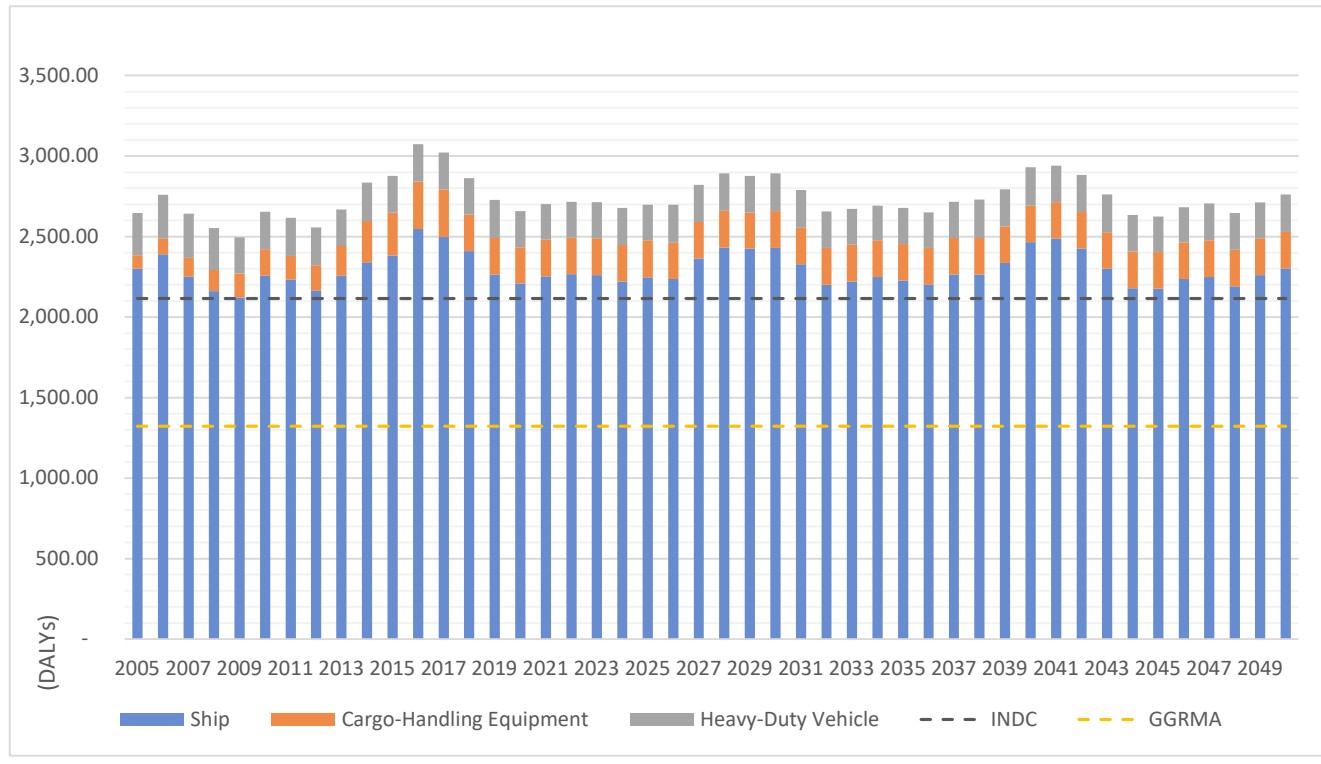

**Figure 4.** External health costs of shipping-related $PM_{2.5}$ emissions (DALYs).

The index of health impact (IHI) is shown in Figure 5. From 2005 to 2050, the average IHI value for shipping-related transportation in Kaohsiung is shown to be 7.21%. The average value of ship emissions in the Port of Kaohsiung is 6.04%, and the emissions from heavy-duty vehicles and cargo-handling equipment in the port is 0.61% and 0.56%, respectively. The highest IHI value from 2005 to 2050 was found to be in 2016, with a value of 8.09%. Since ships are shown to be the main emission source of $PM_{2.5}$ in this research, fluctuations in ship IHI values are highly related to IHI values as a whole. For the years covered by two of the scenarios analyzed in this study, as shown in Figure 5, the IHI value in 2030 is projected to be 7.62%, while that in 2050 is projected to be 7.27%. Both are higher than the IHI in 2005, which was 6.97%.

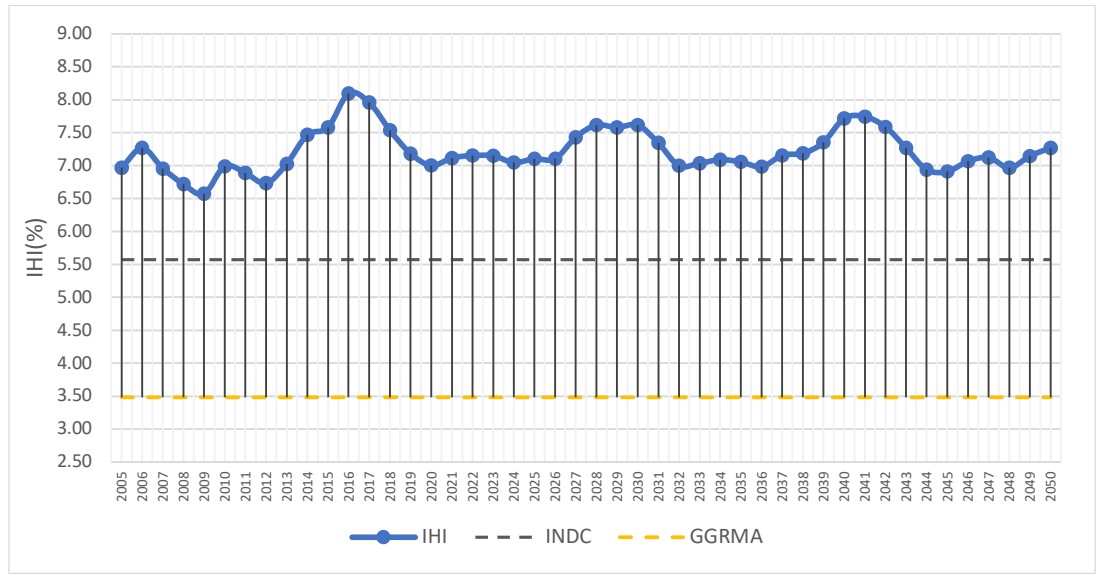

**Figure 5.** Index of health impact of shipping-related PM$_{2.5}$ emissions (%).

In terms of environmental damage, as shown in Figure 6, from 2005 to 2050, the average external environmental cost is projected to be approximately USD 1797.99 million. In the BAU scenarios, the environmental damage cost is projected to be USD 1898.48 million in 2030, which is higher than in 2005. The external environmental cost in 2050 is predicted to be USD 1812.30 million although there is a declining trend when compared with 2030. Nevertheless, the value is still higher than in 2005.

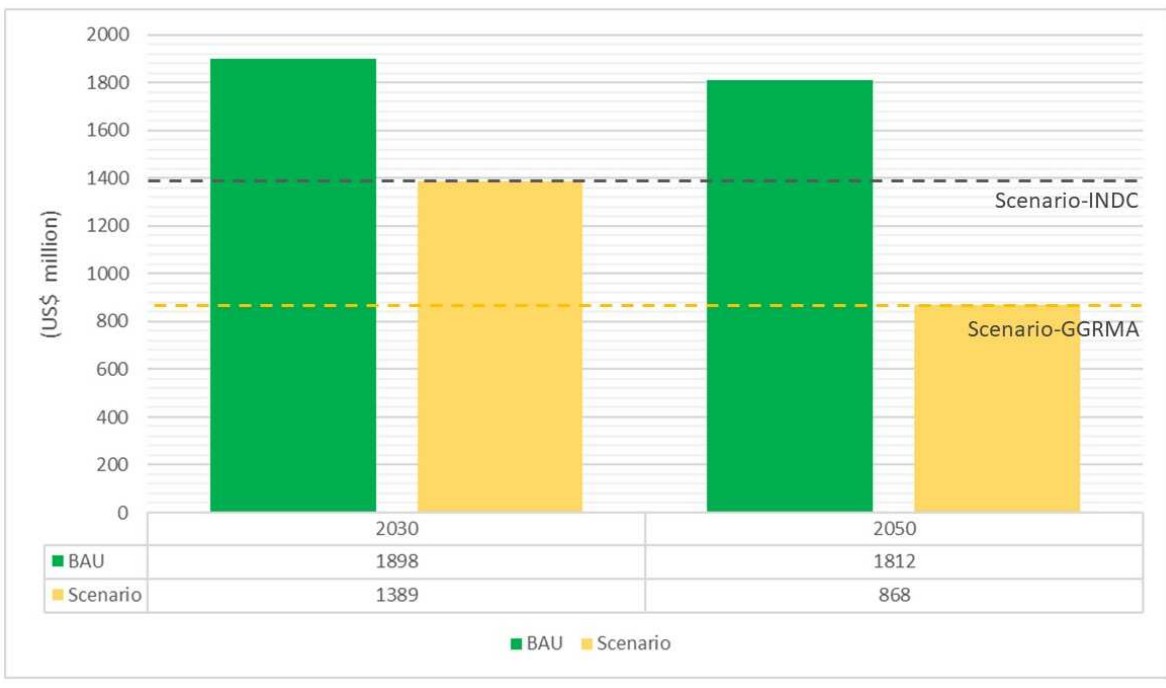

**Figure 6.** External environmental costs of shipping-related PM$_{2.5}$ emissions (million USD).

Scenario-INDC and Scenario-GGRMA in this study estimated what would happen if the regulations in these two acts were enforced. Under the control of the Intended Nationally Determined Contributions, the external health cost will be reduced to 2115.93 DALYs by 2030. This is a reduction of approximately 773.98 DALYs from the BAU scenario. Furthermore, the IHI value is 5.57%, which is 2.05% lower than that of BAU-2030. The external

environmental cost of $PM_{2.5}$ emissions for shipping-related transportation in Scenario-INDC is projected to be USD 1389.07 million, which is a decrease of USD 509.41 million compared to BAU-2030. The effect of the Greenhouse Gas Reduction and Management Act for 2050 is shown in Figure 6, which shows that there would be a 50% decline compared to 2005.

## 4. Conclusions and Policy Implications

*Conclusions*

In this study, an activity-based emissions inventory model was used for ports to assess the $PM_{2.5}$ emissions due to shipping-related transportation in Kaohsiung, Taiwan. The health and environmental impacts of air pollutants on the local area were also assessed. Furthermore, emissions and external environmental costs were calculated in scenarios where the Intended Nationally Determined Contribution Act and the Greenhouse Gas Reduction and Management Act were considered to determine how much these regulations would decrease emissions in 2030 and 2050 when compared to a business as usual scenario. A summary of this study is as follows:

1.　The relative percentages of ships, cargo-handling equipment, and heavy-duty vehicles included in this study were 87%, 6%, and 7%, respectively. The type of ships producing the most $PM_{2.5}$ particles were found to be container ships, which produce 71% of total emissions. The other producers of shipping emissions were bulk ships (15%), tankers (10%), and fishing ships (4%). The types of activities producing emissions include hotelling, which produced 74% of the emissions, followed by cruising (24%) and maneuvering (2%).

2.　From 2005 to 2017, shipping-related transportation operations in Kaohsiung produced 3245.03 tons of $PM_{2.5}$ particles emissions, with USD 1898.42 million in environmental costs, an external health cost of 2722.58 DALYs, and an average health impact index of 7.17% annually, which means that 7.17% of the $PM_{2.5}$-related diseases in Kaohsiung could be due to shipping-related transportation.

3.　For Scenario-INDC in 2030, when compared to the BAU scenario, particle emissions were projected to decrease by 924.88 tons of $PM_{2.5}$; environmental losses were projected to decrease by USD 509.41 million per year; external health costs were projected to decrease by 773.98 DALYs, and the IHI value was projected to decrease by 2.05%, which means that 2.05% of $PM_{2.5}$-related diseases would be avoided, including stroke, ischemic heart disease, lung cancer, chronic obstructive pulmonary disease, acute lower respiratory infections, etc.

4.　In terms of Scenario-GGRMA in 2050, when compared to the BAU, particle emissions were projected to decrease by 2069.46 tons of $PM_{2.5}$; environmental costs were projected to decrease by USD 1139.84 million per year; external health costs were projected to decrease by 1736.28 DALYs, and the IHI value was projected to decrease by 4.58%, which means 4.58% of $PM_{2.5}$-related diseases would be avoided.

In conclusion, the application of the regulations included in INDC (the Intended Nationally Determined Contribution Act) and GGRMA (the Greenhouse Gas Reduction and Management Act) could decrease external environmental costs related to $PM_{2.5}$ particle emissions, including health costs, the percentage of related diseases, and costs related to environmental damage. To reduce emissions and external costs, it is recommended that ships, which are the largest source of emissions, should be the main focus of efforts to mitigate emissions. Dong and Pan [47] also stated that $CO_2$ would significantly decrease the emission of $PM_{2.5}$. Various strategies to reduce the emissions of cargo-handling equipment and heavy-duty vehicles should also be considered. Among several researchers who recommend policies to mitigate emissions, Chand and Jhang [48] suggested that reducing speed and fuel transfer on vessels could reduce greenhouse gas emissions. A carbon allowance allocation could be a policy by which to mitigate vessel carbon emissions [33]. Shore power systems could save energy and reduce carbon emissions [49–51]. For cargo-handling equipment and heavy-duty vehicles, a better alternative fuel is hydrogen, followed by

electricity and liquefied natural gas [52]. If PM$_{2.5}$ particle emissions could be reduced, the health and environmental costs caused by the shipping industry in Kaohsiung would decline significantly.

**Author Contributions:** Formal analysis, C.-C.C.; resources, P.-C.H.; writing—original draft, Y.-W.C. All authors have read and agreed to the published version of the manuscript.

**Funding:** This study was financially supported by the Ministry of Science and Technology, Taiwan, for providing partial funding to support this study under contract number MOST 110-2410-H-006-104.

**Institutional Review Board Statement:** Not applicable.

**Informed Consent Statement:** Not applicable.

**Data Availability Statement:** Data underlying the results presented in this paper are not publicly available at this time due to being part of ongoing work, but may be obtained from the authors later upon reasonable request.

**Conflicts of Interest:** The authors declare no conflict of interest.

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
