# Peer review of "Effects of the INDC and GGRMA Regulations on the Impact of PM2.5 Particle Emissions on Maritime Ports: A Study of Human Health and Environmental Costs"

_sustainability, doi:10.3390/su14106133_

Round 1

Reviewer 1 Report

The authors suggested that the emission inventory is an activity-based bottom-up EI. However, the model lacks resolution and it seems that it is only a collection of multiple EIs available for trucking or shipping. The emission rates are simply a multiplication of EFs to activity presumably (not sure) taken from cited EIs.

Lots of details are missing. Time resolution and spatial resolution are not defined and described. There is no effect of seasonality. The population exposed to PM2.5 has not been identified. The scenarios are now known. How does the regulation decrease PM2.5? Is it by limiting activity? Emission factor? How? is it a technology change? Fuel? Any electrification? Fuel quality? What about current baseline technologies for both trucks and ships? Is it regulated? How? what are the emission control devices? Age distribution? Without having those details, it is hard to understand the emission inventory.

The inventory (baseline) has not been verified by any means. Is it related to concentration in anyways? If so, is there any comparison with ground observation? Has it been compared with atop-down inventory, from fuel consumption or overall energy demand?

How does GHG regulation improve PM2.5? Where is the relevance?

The environmental and health cost is even more simplistic via equations 11 to 13.

Author Response

Thanks for the reviewer 1's comments. All the comments reply as attached file.

Reviewer 2 Report

In this paper the authors estimate the PM2.5 emissions from ships, cargo handling equipment, and heavy-duty vehicles in the Port of Kaohsiung, Taiwan based on the available date for the period 2005-2017. In order to quantify the impact of PM2.5 emissions external health costs, the index of health impact, and external environmental costs were assessed. In addition, the estimation of the health and environmental costs from the PM2.5 emissions from shipping related transportation in Kaohsiung in 2030 and 2050 in a business-as-usual scenario is presented following with the projection of the health and environmental costs if the regulations (emission reduction) specified in the local legislations (the Intended Nationally Determined Contribution Act (INDC, 2015) and the Green house Gas Reduction and Management Act) are achieved.

Generally, the topic is very interesting having in mind growing air pollution problem. From the scientific point of view there is no new findings, but the specific data related to the ship transportation emission and presented results could be useful for future similar studies in terms of comparison.

In the introduction part the goal of the study is clearly defined and most important previous studies and challenges have been analyzed.  The methodology is described with reasonably details and the obtained conclusions are rationally supported by the results obtained.

There are a few shortcomings that should be addressed prior to publication. Please find below several suggestions that can be used for improving the manuscript.

General comment:

I would suggest to include Supplementary material containing basic information (descriptive) on the data used.

There are many acronyms in the text and sometimes it could be hard for readers to follow it. Adding some kind of “the list of abbreviations” would be helpful. Also, there are many redundant full names used (for example the Intended Nationally Determined Contribution) which can be omitted.

Typing errors and technical issues

Line 27: Please use PM2.5 – 2.5 in subscript

Many acronyms are used but associated full name should be included at the first mention (for example DALY, EEA, IEA…)

Through the text

Lines 95-96:”… the Intended Nationally Determined Contribution (INDC) and the Greenhouse Gas Reduction and Management Act (GGRMA),…”  Please use only INDC and GGRMA instead, since it has already defined previously

Line 169” … Kyu et al. (2018), and, and the model is…”.  Please delete “and”

Equations 11 and 12 are the same, one of them can be omitted due to the shortening

Author Response

Thanks for the reviewer 2's comments. All the comments reply as attached file.

Reviewer 3 Report

The presented study's aim was to estimate the PM2.5 particle emissions from ships, cargo handling equipment, and heavy-duty vehicles in the Port of Kaohsiung, Taiwan. 
The idea tends to be interesting and valuable. What is missed in the article is its classical structure of it. In my opinion, the literature review is the section that is obligatory for the scientific paper. 

Author Response

Thanks for the reviewer 3's comments. All the comments reply as attached file.

Round 2

Reviewer 3 Report

I feel satisfied with the improvements done by the aurhors.